# *Shaka Goichidaiki Zue*: Vernacularization and Visualization of Buddha's Biography in Nineteenth-Century Japan

**Wei Xiang**

School of Foreign Languages, Peking University, Beijing 100871, China; xiangwei_koui@pku.edu.cn

**Abstract:** Since the appearance of Buddha, texts and images depicting his life have circulated across Eurasia, serving as significant mediums for disseminating Buddhist ideology. Japan has historically been influenced by the canon of Chinese Buddhism while concurrently striving to promote the indigenization of Buddhism. This endeavor reached its peak during the Edo period, notably exemplified in the *Shaka goichidaiki zue*, illustrated by the world-renowned artist Hokusai Katsushika. Originating from Buddhist believers, it presents an adaptation based on the socio-historical context of pre-modern Japan, particularly manifesting evident shifts in emphasizing royal authority, the salvation of females, and ethical relationships. Entering the Meiji era, this pre-modern illustrated manuscript underwent repeated printing, playing an important role in the modernization of Buddhism.

**Keywords:** *Shaka goichidaiki zue*; Shakyamuni; Hokusai Katsushika; nineteenth-century Japan

## 1. Introduction

The Shakyamuni's life has shared common narrative structures and iconographical conventions across Asia for centuries. The absence of a singular, authoritative textual source for the Buddha's life has led visual narratives to historically center around various canonical episodes. These episodes encompass key events, ranging from the Buddha's birth, ascetic practices, to his enlightenment and Nirvana. While these narratives eventually coalesced into standardized templates, typically featuring Hassō 八相, namely, eight major episodes, they also exhibited creative flexibility. This flexibility manifested through the addition of new episodes or the extension of the narrative, allowing for contextual adaptation.

In Japan, narratives about Shakyamuni evolved from collections of individual short stories about him, such as the *Sanbō-e* 三宝絵 and *Konjaku monogatarishū* 今昔物語集 in the Heian era, to independent stories, such as the *Shaka no honji* 釈迦の本地 and *Shaka goichidaiki zue* 釈迦御一代記図会 in the Edo period. These narratives were not only read as textual content but also orally transmitted during religious ceremonies, such as the Buddha's birth celebration. Additionally, they were expressed through various forms, including statues, wall paintings, handscrolls, folding screens, and picture books. For instance, there is a seventh-century Nirvana statue in the five-storied pagoda at Horyuji Temple. The handscroll *E-inga-kyō* 絵因果経 in the eighth century, albeit incomplete, vividly depicts stories such as the travels of Shakyamuni through the four gates. Diverse expressive media intersected, creating a complex and layered composite. Visual culture, especially in the form of painting, played a significant role, with vernacular Buddha's biographies.

Kurobe Michiyoshi and Komine Kazuaki have conducted pioneering studies on Japanese Buddhist literature, highlighting a historical lack of attention to Edo-period Buddhist narratives in academia (Kurobe 1989; Komine 2017), while the actuality reveals that Japanese Buddhist narratives could not break away from Chinese norms and influence and display unique characteristics until the Edo period. Micah Auerback observed the canon and creation in the making of a Japanese Buddha and attempted to categorize Japanese

Buddhist narratives into five types, namely the Buddha as Preceptor, Local Hero, Exemplar, Fraud, or Character (Auerback 2016). The classification of Buddha's narratives seems mostly appropriate, considering both the chronological development and typology based on content. However, the second and third types are mainly relevant to Edo-period Buddha's narratives, with a somewhat rigid approach using religious stance as a criterion for classification. For example, Maika points out that *Shaka hassō yamato bunko* 釈迦八相倭文庫 by Mantei Ōga 万亭応賀 represents a narrative toward storytelling and entertainment, authored by laypeople. In contrast, *Miyo no hikari* 三世の光 is categorized as a work written by a disciple, Ritsu-shū nun Kōgetsu Sōgi 皓月宗顗, questioning the essence of Buddhism and expressing a desire to depict Buddha as a model. In fact, various Buddha narratives in pre-modern Japan are quite complex and layered and it is not advisable to approach the discussion using a binary opposition. An example illustrating this complexity is the *Shaka goichidaiki zue.* Despite originating from Buddhist believers, this work strikingly integrates this-worldly elements with other-worldly components, making it a culmination of Buddha narratives in the Edo period.

Largely overlooked by scholars of Japanese Buddhism, there is little related research for this work. It appears that Sekiguchi Masayuki's work is the only prior study focused on the *Shaka goichidaiki zue*. He emphasizes the significance of this work as a representative example of Buddhist visual expression in pre-modern Japan (Sekiguchi 1989). Nevertheless, art historian Sekiguchi's study did not extensively explore the relationship between text and image or delve into the religious studies aspect. In recent years, Sueki Fumihiko has been trying to reveal the characteristics of Japanese pre-modern Buddhism through popular works, such as Buddha's biographies (Sueki 2018, 2022), but he has not specifically discussed the *Shaka goichidaiki zue*, leaving room for further investigation.

Building upon prior research, this article aims to analyze the vernacularization and visualization of the *Shaka goichidaiki zue* and discuss its influence and significance within the context of 19th-century Buddhist thought. Although the academic tendency to separate pre-modern and modern Japan is present, this article emphasizes the inheritance of Buddhist cultural heritage between the Edo period and the Meiji period through an examination of the life cycle of a famous book—the production and reprints of the *Shaka goichidaiki zue* in the 19th century.

## 2. Production of the *Shaka Goichidaiki Zue*

The *Shaka goichidaiki zue* is the illustrated record of Shakyamuni's life, published in 1845, which consists of six volumes.[1] The entire work is composed of 55 episodes compiled by Yamada Isai 山田意斎 and 35 monochrome illustrations crafted by Katsushika Hokusai 葛飾北斎. This book has three publishers: Echigoya Jihei 越後屋治兵衛 in Kyoto, Yamashiroya Sahei 山城屋佐兵衛 in Edo, and Kawachiya Mohei 河内屋茂兵衛 in Osaka, indicating that it was prevalent during that time.

In 1921, Furuya Tomoyoshi 古谷知新, a renowned literary researcher in the Taishō Era, was compiling the *Nihon Rekishi Zue* 日本歴史図会, and he once said:

> In Japan, there is no shortage of Shisho 史書 (historical books), but historical books appealing to the general public are rare. While there are accurate historical records and official histories, they tend to be generally dry and uninteresting. On the other hand, entertaining narratives often found in Kōdan 講談 (oral storytelling) are frequently overly fantastical. However, there exists a kind of illustrated historical book that is not only easily comprehensible to the general audience but also adhering to historical facts. This form emerged towards the end of the Tokugawa period, namely Zue. (Furuya 1921a, p. 1)

Furuya perceptively pointed out that the "zue" genre balanced the accuracy of historical records with the entertainment of oral storytelling. However, he overlooked the religious significance inherent within the "zue" itself. Despite the independent emergence of "zue" as a literary genre during the Edo period, its usage can be traced back to the Heian period. For instance, in the *Konjaku monogatarishū* (Volume 6, Episode 16), the passage men-

tions that a monk in a Chinese temple, upon encountering a statue of Amitabha Buddha, proceeded to depict it in a painting ("zue"). Subsequently, he engaged in daily worship, ultimately attaining rebirth in the Pure Land (Komine 1999, p. 44). Additionally, the *Tōdaiji Zokuyōroku* 東大寺続要録" (circa 1281–1300) references the production of new images ("zue") of Bodhisattvas for religious purposes, followed by the recitation of the Heart Sutra 心経. This indicates that the term "zue" primarily related to religious contexts, generally referring to the images or drawings of Buddha and Bodhisattvas displayed during rituals or offerings. Essentially, "zue" often served as a visual and accessible representation of religious faith, usually depicting the imagination of the other world.

Then, Furuya included the *Shaka goichidaiki zue* in the eighth volume of the series. He explained the reason in the preface: "The *Shaka goichidaiki zue* in 1845 depicts the life of Shakyamuni in an easy-to-understand way. There is the *Shaka hassō monogatari* 釈迦八相物語 (1666) preceding it, and later there are works like the *Shaka hassō wa bunko* 釈迦八相倭文庫 (1851–1859). However, in terms of text and illustrations, there is no comparison to this book" (Furuya 1921b, p. 1). Furuya Tomoyoshi highly praised the excellence of Shaka goichidaiki zue in both text and image, considering it far superior to contemporary works. His intuitive perception is correct, but he did not provide further explanation on this matter. In fact, the Shaka hassō monogatari is a famous work with a broad audience, and it was even adapted by the Japanese esteemed playwright Chikamatsu Monzaemon 近松門左衛門, into the *Shaka nyorai tanjōe* 釈迦如来誕生会. Although the author of the Shaka hassō wa bunko remains relatively obscure, the illustrations are the work of prominent Ukiyo-e artists from the Utagawa school, such as Utagawa Toyokuni II 二代目歌川豊国 and Utagawa Kunisada II 二代目歌川国貞, showcasing artistic solid expression. Therefore, it is of significance to explore why the *Shaka goichidaiki zue* outshines others. This article argues that the reason for the tremendous success and influence of the Shaka goichidaiki zue ultimately lies in the fact that in the process of popularization, this work did not completely forsake its religious nature and avoided degradation into a mere object of enjoyment. Instead, it effectively achieved a balance between this-worldly and other-worldly elements. The following will discuss the participants of the book.

As a globally recognized artist, Katsushika Hokusai's extensive work has been subject to substantial scholarly exploration. Scholars typically categorize his career into six distinct periods based on different pseudonyms, namely the "Shunrō period 春朗期" (20–34 years old), "Sōri period 宗理期" (35–45 years old), "Hokusai period 北斎期" (46–50 years old), "Taito period 戴斗期" (51–60 years old), "Iitsu period 為一期" (61–74 years old), and the "Gakyōrōjin Manji period 画狂老人卍期" (75–90 years old). It is noteworthy that the pseudonym "Hokusai" from the third phase became so renowned that, to avoid audience disconnection, for commercial promotional purposes, Hokusai continued to use it alongside other pseudonyms in subsequent periods. Scholars have observed that each change in pseudonyms signifies a renewal in subject matter or artistic style. For instance, the renowned work "The Great Wave off Kanagawa" was created during the "Iitsu period", during which Hokusai passionately embraced landscape prints, yielding significant accomplishments. In contrast, the *Shaka goichidaiki zue* was crafted during the final period, the "Gakyōrōjin Manji period", during which Hokusai shifted his focus from standalone Ukiyo-e to producing illustrations for books. Consequently, scholars traditionally concentrating on Ukiyo-e have paid limited attention to this period. Illustrations in this book are not widely known.

In the fifth year of the Tenbō era (1834), at the age of 75, Hokusai made his final change to his artistic pseudonym, adopting "Gakyōrōjin Manji". The "Manji 卍" symbol represents Buddhist teachings and cosmic harmony. "Gakyōrōjin" translates to "the elderly person mad about painting", reflecting Hokusai's vision of not ceasing his artistic endeavors even in his advanced years. Hokusai candidly expressed when he was 75 years old: "I have been obsessed with copying and drawing since the age of 6. I produced many works (illustrations for books) around the age of 50. However, what I drew before the age of 73 is not worth mentioning (such as the famous 'Thirty-Six Views of Mount Fuji'). It was only

after the age of 73 that I grasped the skeletal structure of birds, animals, insects, fish, and how plants grow (flower and bird paintings). It will be at the age of eighty that I take a step further. I will delve into the mysteries of painting at 90 and become proficient at 100. At 110, my brush could create wonders, every stroke lifelike. Oh, the deity of longevity, I would like you to witness that what I say is not a fabrication" (Iijima 1893, p. 53). Evidently, in his old age, Hokusai harbored a strong determination for longevity and artistic creation. However, behind this aspiration for longevity, there lay an underlying unease about the realities and the afterlife. This period coincided with the Tenpō Famine 天保の大飢饉 and a challenging time for the publishing industry due to the Shogunate's Tenpō Reforms 天保改革, particularly the reinstatement of the prohibition of luxury. Compounded by the gambling addiction of his grandson, Hokusai faced looming bankruptcy. Forced by circumstances, he left Edo and continued his creative pursuits with the sponsorship of the affluent farmer-merchant Takai Kōzan 高井鴻山 in the town of Obuse, Nagano Prefecture.

Notably, Katsushika Hokusai was a devout Nichiren Buddhist (Nagata 2003),[2] yet he scarcely explored Buddhist themes before the age of 75. In the twilight of his life, Hokusai deliberately incorporated conflicting elements, "Kyō 狂" and "Manji 卍", into his pseudonym. He embarked on the creation of Buddhist-themed works, perhaps seeking a balance between artistic pursuits and the realities of life through religious belief. The *Shaka goichidaiki zue* stands out as Hokusai's first venture into Buddhist themes. Following its success, he even produced a biography illustration book titled *Nichiren shōnin ichidaiki zue* 日蓮上人一代図会 (1858) for the founder of the Nichiren school, whom he devoutly followed.

In comparison, there is limited scholarly research on the compiler of the *Shaka goichidaiki zue* currently. Based on scattered biographical information found in works such as *Keisetsu gesakusha kō* 京抵戲作者考 and *Naniwa jinbutsu shi* 浪華人物誌, we glean that the author was known as Yamada Isai, originally named Yamada Keizō 山田桂藏, also known as Yamada Kakashi 山田案山子, and bearing the pseudonym of Kōkado Yatei 好華堂野亭. He excelled in the creation of yomihon, jōruri, and kyōka. In his later years, Yamada Isai shaved his head, assumed the new name Isai 意斎, and expressed, "I am not a priest, not a tea master, not a doctor either. I am just a monk" (Kokusho Kankōkai国書刊行会 1908, p. 277). He did not formally enter the Buddhist priesthood but chose a path of lay practice, making him a distinctive writer. Similar to Hokusai, Isai primarily engaged in historical storytelling until his later years when he ventured into Buddhist works. Isai passed away suddenly the second year after the formal publication of the *Shaka goichidaiki zue*, at the age of 59. His other Buddhist work, *Kannongyō wakun zue* 観音経和訓図会 (Illustrations and Explanations of the Guanyin Sutra), did not see the light until 1849.

Examining the life experiences of Hokusai and Isai, it becomes apparent that, in their later years, facing the inevitability of aging and mortality, both individuals found a certain resonance within themselves. It led to their first and only collaboration, earnestly tackling Buddhist themes that had previously been overlooked. The suffixes "sō 叟" added to their names in the books, namely "Katsushika Hokusai-sō" and "Yamada Isai-sō", indicate a self-deprecating tone, derived from the Chinese character "shou 瘦", denoting a thin and emaciated elderly person, but also carrying an implicit optimistic spirit willing to contend with the aging of the body.

In addition, another significant character played a crucial role in the promotion of this book, namely the renowned Buddhist monk Daikō Sōgen 大綱宗彦 (1772–1860) in the late Edo period. Sōgen served as the abbot of the Daitoku-ji 大徳寺 temple in Kyoto. Sōgen was proficient in poetry and excelled in calligraphy and painting and had extensive interactions with influential figures outside Daitoku-ji. In 1839, at the age of 68, Sōgen specially wrote a preface for the *Shaka goichidaiki zue*, stating:

> Oh, Shakyamuni, your advent into this world is akin to radiance. Your ascetic practices were exceedingly arduous. Your teachings are remarkably profound. If the teachings are not expounded with the skillful use of expedient means, without distinguishing between immediate and gradual methods, how can one effec-

tively liberate and enlighten the world? Though the understanding of worldly beings varies due to their diverse ignorance, their fundamental nature is undivided. As the scripture says, "There is no birth and no death, neither an increase nor a decrease". Thus, it is asserted that the truly supreme path is not difficult to comprehend. For those who read this book, they should carefully ponder why Hokkei expended such immense time and effort willingly. It is no longer necessary for me, a humble and ignorant monk, to dwell on this book.

光哉、世尊出世。難哉、苦行。深哉、説法。不知何設権実、頓漸之法薬、以度一世乎。夫知愚各異、其性無二。経云、不生不滅、不増不減。故曰、至道無難読此書者、見野亭北斎為何物徒労。愚衲紹介、何益之有。 ([Furuya 1921b](#), p. 1)

In fact, Daikō Sōgen adhered to the Rinzai Zen, exhibiting certain doctrinal differences from the adherent of Nichiren Buddhism, Hokusai. However, both individuals set aside sectarian differences, aiming to jointly propagate Buddhism. In the preface, Sōgen first praised the life of Shakyamuni, then quoted the *Heart Sutra* to emphasize the diverse ignorance of worldly people, all possessing Buddha nature equally. Therefore, to save and enlighten the world, the Buddha's teachings necessitate Kaigon kenjitsu 開権顕実, the skillful use of expedient means to reveal the true reality. In this way, the profound teachings of Buddhism become understandable. It is evident that Sōgen did not reject the practice of vulgarizing Buddhist teachings. Instead, he positively affirmed this book from the perspective of broadening accessibility through skillful means and praised the efforts of Isai and Hokusai in disseminating Buddhist teachings.

Isai and Hokusai indeed dedicated considerable effort to this book. Details about Isai's creative process are scant, but recently, Shūgō Asano and Sarah Thompson rediscovered a set of Hokusai's original drawings, *Hokusai sensei gafu* 北斎先生画譜, at the Boston Museum of Fine Arts. During the woodblock printing process in the Edo period, there was a drawing technique known as "block-ready drawing" (Hanshita-e 版下絵). In this method, the artist's final drawing, created with ink lines, was inverted, and placed on a woodblock for carving. The completed carved block was then used by printers to produce ink impressions. However, before reaching this stage, artists often manually sketched the content, creating rough drawings. This set of drawings comprises illustrations for the *Shaka goichidaiki zue*, presented by Hokusai to his pupil Manjirō Hokuga 卍楼北鵞. Subsequently, it was collected by Katsushika Hokusen 葛飾北僊, William Sturgis Bigelow, and ultimately housed in the Boston Museum of Fine Arts ([Thompson 2023](#)).[3]

It is worth noting that Manjirō Hokuga pasted these drawings into an album dated 1835. This indicates that Kokusai commenced the design process at least ten years before the publication of the *Shaka goichidaiki zue*. Hokuga cherished these drawings, presumably attracted by their artistic expression, prompting him to seek Hokusai's work for emulation. Then, what were they seeking to express through this book?

## 3. Reception and Recreation in the *Shaka Goichidaiki* Zue

[Sekiguchi](#) ([1989](#), p. 742) examined the similarities and differences with traditional Buddha's images and pointed out that among the 35 illustrations depicted in the *Shaka goichidaiki zue*, only 12 scenes originate in the former or traditional ones of these subjects. Furthermore, two-thirds of the total illustrations are possibly represented as new iconographies in Shaka's biography. In fact, Hokusai may be inspired by various pictorial resources, including Japanese traditions, although he has a skillful technique and inexhaustible imagination. For instance, Figure [1](#)a depicts Shakyamuni facing the final trial to attain enlightenment, engaging in a dialogue with a demon (revealed as the Buddha Lushena). The grotesque visage of the demon, its single leg lifted off the ground, and the gaze directed downward, among other detailed movements and the overall composition, bear a remarkable resemblance to the crimson-hued demon portrayed in *Hyakki yagyō emaki* 百鬼夜行絵巻. It indicates that Hokusai endeavored to depict other-worldly beings with a traditional Japanese aesthetic. In addition to the visual elements, there is a noticeable

inclination toward indigenization in the textual narrative of this book. The adaptation and recreation of Buddha's story are primarily manifested in three aspects, as below.

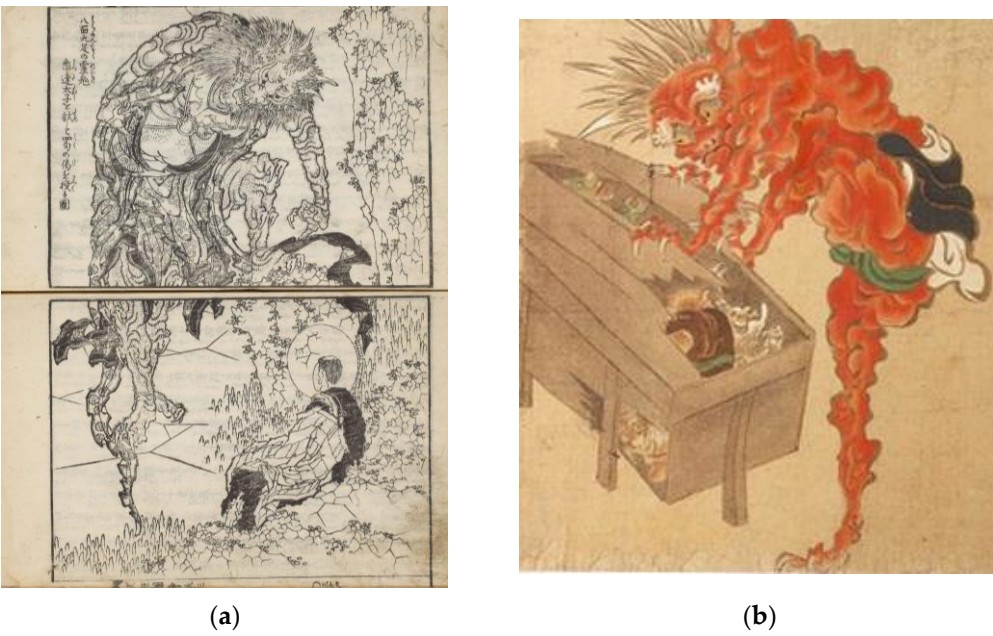

| (**a**) | (**b**) |

**Figure 1.** (**a**) *Shaka goichidaiki zue* Volume 3. Caption: "The demon with eight faces and nine feet tests Prince Siddhartha, and he receives the four-line verse". © Kobe University Library, Japan. (**b**) *Hyakki yagyō emaki*. A crimson-hued demon discovered the minor demons concealed within the wooden cabinet (Komatsu 1979, p. 82).

### 3.1. Concordance of Royal Authority and Buddhism

This book begins with the ascension of Sudhdhodhana, the father of Shakyamuni. It is said that the country of Tenjiku consists of east, south, west, north, and central realms. The central kingdom, Kapilavastu, is ruled by the Wheel-Turning King, who governs all the realms. Subsequently, the narrative traces the lineage of the Wheel-Turning Kings. The twenty-sixth in line is King Sihahanu, who has four sons. According to Buddhist literature, such as the *Shiju hyaku inenshū* 私聚百因縁集 and *Sangoku ddenki* 三国伝記, it is stated that King Sudhdhodhana is the eighty-four thousandth two hundred and tenth generation from the Wheel-Turning King. However, in this narrative, King Sihahanu's lineage is condensed to the 26th generation, seemingly to enhance a sense of realism for a better understanding in Japan and to avoid an overly extensive temporal narrative or mythical distance. As we know, despite the Japanese emperors claiming an uninterrupted lineage, even the final Emperor Kōmei of the Edo period was only the one hundred and twenty-first generation.

Furthermore, this work envisages an idealized realm of peace and prosperity under King Sihahanu's rule. King Sihahanu reigned for fifty years, gathered all his ministers in the court, abdicated in favor of Prince Sudhdhodhana before them, and appointed the other three princes as kings of smaller territories. He then retired to a hermitage to live out his remaining years. King Sudhdhodhana was also virtuous and dedicated to good governance, resulting in a prosperous and harmonious nation. The people lived in peace and abundance, experiencing favorable weather conditions, and all citizens were regarded as brothers (Furuya 1921b, p. 12). The book extensively extols this flourishing era, praising it as an era of clairvoyance and unparalleled holy teachings, which echoes the era where the author lived. The Japanese have long held a belief in the "word spirit" (Kotodama 言霊), whereby the conviction that positive blessings can influence the present world is ingrained. Consequently, individuals express their admiration for the present world through forms such as classical waka poetry and traditional Noh drama. However, the notion of praising

the present world in the context of religious narratives is uncommon, especially considering that Buddhist tales traditionally focus on the stories of past lives and emphasize the ephemeral nature of the present life. The deviation in this narrative is evident by initiating the Buddhist tale with a celebration of the present world, distinct from typical Buddhist storytelling.

Within this Buddhist biography, royal authority is magnified and narratives pertaining to this-worldly elements are incorporated. For instance, Maha Maya's father, Suprabuddha, is portrayed as a military general and minister, fully depicted within the context of Japan's imperial dynasties. Moreover, according to the *Sutra of the Collected Stories of the Buddha's Deeds in Past Lives* 仏本行集経, the Shakyas were originally a small clan and were said to have been destroyed by an attack from the Kosala kingdom during Buddha's lifetime. However, in the *Shaka goichidaiki zue*, King Sudhdhodhana is depicted as a Wheel-Turning King who ruled over all the five Tenjiku realms. It is asserted that Buddha was born into a royal family boasting immense power.

The first illustration in the work depicts envoys from various countries arriving at the capital of Kapilavastu to pay homage to King Sudhdhodhana, who has just ascended the throne. This prostration scene is echoed after the birth of Prince Siddhartha. After Prince Siddhartha's birth, this book describes the scene where celestial dragons pour warm and cold water from the heavens to bless him. This episode is seldom depicted in previous Buddha's biographies. However, this narrative is elaborately portrayed across two pages, emphasizing the unique composition. As Sekiguchi pointed out, the intertwining bodies of the two dragons, reminiscent of ceiling paintings in pre-modern Japan, create a complex visual narrative (Sekiguchi 1989, p. 732). The viewer is compelled to look upward at the dragons and heavenly beings in Figure 2, only to be guided back to focus on Prince Siddhartha. The composition is distinctive, placing Prince Siddhartha and the dragons on the left side of the composition, with various figures and heavenly beings added, dividing the page horizontally. The upper page (right page) depicts the heavenly beings gazing at Prince Siddhartha, while the lower page shows the people on the ground kneeling before him.

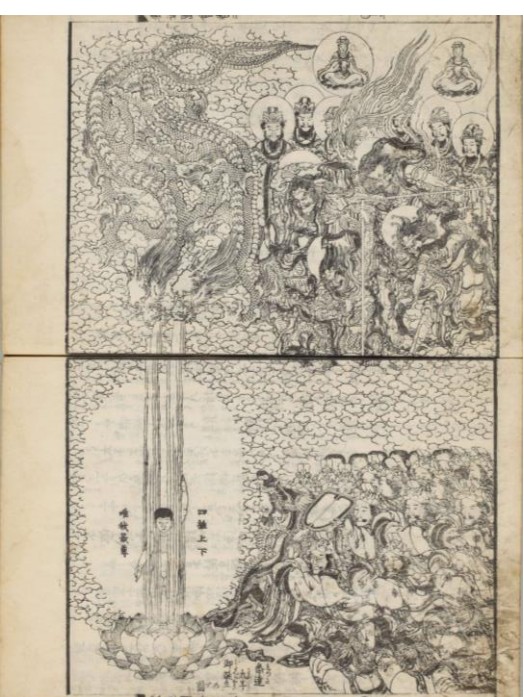

**Figure 2.** *Shaka goichidaiki zue* Volume 2. Caption: "In the four dimensions, I stand alone with unparalleled authority". Image cited from: https://kokusho.nijl.ac.jp/biblio/100389889/55?ln=ja (accessed on 3 January 2024) © Kobe University Library, Japan.

The birth of Buddha, as portrayed in the *Sutra of the Cause and Effect* 因果経, is fantastical and beautiful, demonstrating the joy of all beings upon encountering Buddha's birth through sound, fragrance, and sight. Lotuses of the seven treasures emerge under the tree, the heavenly dragons perform music, and fragrances are emitted while the gods scatter jewels. The depiction is characterized by solemn religiousness. However, the illustration in the *Shaka goichidaiki zue* combines this-worldly and other-worldly elements in a coexistent and harmonious manner. The dual nature of the expectations for Prince Siddhartha is evident: heavenly beings above hope for him to become an enlightened sage, while the people below desire him to be a wise king who will govern the world in the future.

Although Prince Siddhartha eventually embarks on the path of enlightenment, foregoing the throne and causing distress to King Sudhdhodhana, the narrative sets up a foreshadowing. When Prince Siddhartha resolves to renounce worldly life, this book deliberately mentions, "On this day, the Lady Kōyō gave birth to a prince. King Sudhdhodhana was so delighted, and this prince was respectfully named Prince Nanda later" (Furuya 1921b, p. 101). The birth of another prince introduced a turning point in the succession issue of the Shakya clan. It is commonly believed that the mother of Prince Nanda is Gautami, but here, there is a deliberate deviation from this convention. Instead, a new character not found in Buddhist scriptures is introduced. Her name, Lady Kōyō 好容夫人, literally means a beautiful appearance. According to the narrative, Lady Kōyō is one of King Sudhdhodhana's consorts, thereby maintaining the purity of the royal lineage. Eventually, Prince Nanda successfully inherits King Sudhdhodhana's throne. On the day of his ascension, people gather to bow before King Nanda, chanting "*Banzai* 万歳 (to live ten thousand years)" in unison.

After resolving the issue of succession, another crucial matter arises: the extinction of the Shakya clan. According to the scriptures, when Vidudabha hears of his father's death, he immediately declares himself king and leads his forces to attack the kingdom of Kapilavastu. However, the *Shaka goichidaiki zue* does not place this conflict in Kapilavastu 迦毘羅 but in the Sakiya Kingdom 舎夷国. In fact, Kapilavastu is the transcription from Sanskrit, while Sakiya is an interpretive translation from Pali, which carries the meaning of a holy or sacred person. Both of them indicate the same country historically. However, the *Shaka goichidaiki zue* treats them as distinct nations, with Sakiya being the territory of King Amritodana, and it is Sakiya that suffered an unfortunate fate from King Vidudabha. Thus, this adjustment in the book implies the continued existence of the Kapilavastu and the Shakya clan.

Examining this narrative alongside the segments that draw parallels between Japanese customs and events, it is evident that Japan's correspondence with Tenjiku has persisted. For instance, following the description of the auspicious events surrounding the birth of Prince Siddhartha, the narrative seamlessly transitions to a mention of the contemporary Japanese custom of adorning temple roofs with flowers during the Buddha bathing ceremony, stating, "Nowadays, the roofs of temples in our imperial country, as well as the shrines for Buddha bathing, are adorned with flowers. It is a custom that has persisted since that time. Similarly, in the houses of Kyoto, azaleas are inserted on the eighth day of April" (Furuya 1921b, p. 58). In this context, the *Shaka goichidaiki zue* strategically emphasizes the lineage and royal authority of King Sudhdhodhana, aiming to evoke associations with the traditions of the Japanese imperial lineage, thereby establishing a certain resonance for the reader.

### 3.2. Female's Jealousy and Salvation

As previously mentioned, this book emphasizes that the mother of Prince Nanda is not Gautami. It is conceivable that the character of Gautami undergoes a noticeable transformation in the *Shaka goichidaiki zue*, initially portrayed as a villain. According to the plot, Maha Maya and Gautami are two princesses and simultaneously become queens, leading to a rivalry for the favor of King Suddhodana. Maya becomes pregnant, and the king, overjoyed, appoints her as the chief queen in the imperial palace. Gautami is so jealous that

she transforms into a snake-like entity. She resorts to poisoning and casting curses upon Maya. However, Gautami finally repents and assumes the role of a nurturing mother to Shakyamuni, with Maya's untimely death one week after giving birth to Shakyamuni. This narrative, unique in its approach among Buddhist scriptures and previous literature, raises intriguing reflections.

The accompanying illustration (Figure 3), consisting of two pages, establishes a stark contrast between good and evil. The right-side illustration depicts the canonical episode of Buddha's conception, featuring Maya's prophetic dream of pregnancy. Conversely, the left-side illustration reimagines Maya's sister, traditionally portrayed as a benevolent aunt, as a jealous antagonist who transforms into a giant snake.

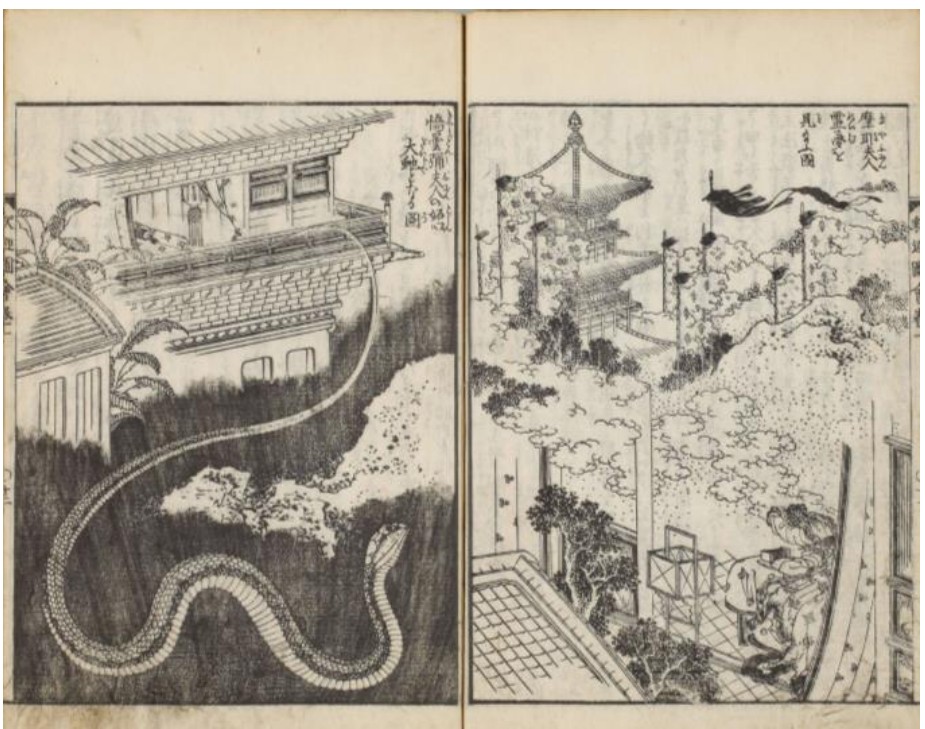

**Figure 3.** *Shaka goichidaiki zue* Volume 1. Caption: "The envious heart of Lady Gautami transforms her into a giant serpent (**Left** image). Maha Maya experiences a prophetic dream (**Right** image)". Image cited from: https://kokusho.nijl.ac.jp/biblio/100389889/24?ln=ja (accessed on 3 January 2024) © Kobe University Library, Japan.

Narratives before the Edo period, such as Buddhist scriptures and *Konjaku monogatarishū*, revolve around the harmonious relationship between Maya, portrayed as the elder sister, and her younger sister Gautami, entrusted with the care of Shakyamuni after Maya's death. In contrast, the *Shaka goichidaiki zue* focuses on the conflict between Maya and Gautami. Gautami assumes the role of the elder sister, while Maya is depicted as the younger and relatively weaker figure. The storyline of Gautami becoming jealous of her sister Maya is not an original creation of Isai and Hokusai, but can be traced back to earlier works, such as the *Shaka hassō monogatari* and the *Shaka nyorai tanjōe*. Regarding the reversal of the sisterly relationship between Gautami and Maya, Fujio (1995) suggests, "Even at the expense of deviating from the Buddhist narrative, the decision to portray Gautami as the elder sister might have been a deliberate narrative choice to enhance realism, as it reflects the anger of an elder sister who, despite being the elder, had her position as the chief queen taken by her younger sister, resorting to the means of subduing that anger".

While the plotline of Gautami's jealousy and attempts to harm Maya is not original from Isai and Hokusai, the reimagining of her as a serpent adds a new dimension. Bryce Heatherly pointed out that this narrative element may be borrowed from the story of Xi Hui

郗徽, wife of Emperor Wu of the Chinese Liang Dynasty. She is said to have transformed into a serpent due to her extreme jealousy in the ritual text *Merciful Penitence of the Ritual Area* (慈悲道場懺法) and other Buddhist proselytizing literature.[4] While this perspective is insightful, there remains a certain degree of detachment from the text, and it lacks sufficient alignment. In comparison to the story of Xi Hui in China, Japanese classical literary traditions bear closer resemblance to the *Shaka goichidaiki zue*.

First and foremost, the theme of rivalry among females, commonly known as Kisaki arasoi 后争, has persisted in Japanese classics, especially *The Tale of Genji*. For instance, Lady Kiritsubo 桐壺 (Genji's birth mother) faced jealousy and curses from the Empress, leading to her early demise. Lady Aoi 葵上, Genji's lawful wife, was possessed by a ghost (the lingering spirit of Lady Rokujō 六条御息所生霊) during her pregnancy. Second, the notion of women transforming into serpents due to jealousy finds resonance in the popular Edo-period legend of Dōjōji Temple 道成寺. This narrative revolves around a woman named Kiyohime 清姫, who was betrayed by a man she admired, and transformed into a serpent seeking revenge, ultimately attaining salvation with the help of the monks transcribing and venerating the *Lotus Sutra* at Dōjōji Temple. Hence, it becomes imperative to focus on how Gautami attains salvation after transforming into a serpent, a facet overlooked by previous studies. . The *Shaka goichidaiki zue* emphasizes that females must be cautious about three aspects (三怠), with jealousy ranking first. This is because the competition for favor arises from greed, and the inner self may be consumed by the fire of anger and resentment (Furuya 1921b, p. 96).[5] Maya's three-year pregnancy is attributed to Gautami's curse, as explained by Shakyamuni in a dream, delaying his birth to three years, while the *Sutra of the Cause and Effect* document portrays a gestation period of 10 months. Gautami, after years of jealousy, experiences a sudden change of heart during a flower banquet at Lumbini Garden. Feeling the awe-inspiring presence of the Tathagata and its radiance, she suddenly overcomes her intense jealousy, regretting making her sister endure the pain of a three-year pregnancy. At this moment, all her sins vanish instantly, allowing Shakyamuni's smooth birth. Gautami's prolonged jealousy dissipating in a single event seems contrary to common understanding but parallels the salvation of Kiyohime at Dōjōji Temple, portraying the compelling influence of Buddhism.

After Maya's death, Gautami raises Shakyamuni and becomes the first nun, called "Great Lover of the Path" (Daiaidō 大愛道) in the *Shaka goichidaiki zue*. This fact, commonly found in Buddhist texts, emphasizes the urgency for Gautami's early salvation. Conversely, Chikamatsu's work, the *Shaka nyorai tanjōe*, overlooks this aspect, depicting Gautami plotting against Maya and Shakyamuni for an extended period. Gautami's salvation is delayed until the Buddha's final sermon. Isai's adjustments and handling of the narrative align with the fundamental facts of the Buddha's life, promoting the idea of "attaining enlightenment in a single thought". Simultaneously, it integrates elements from Japan's literary tradition, making it an intriguing and multifaceted narrative.

*3.3. Emphasis on Ethical Relationships*

Throughout this work, the word *On* 恩 (grace) appears most frequently, occurring 88 times. Shakyamuni himself is also referred to as the Great Master of Grace (Daion kyōshu 大恩教主) in the style of Chinese tradition "Embroidered Portraits" (Xiuxiang 繡像) at the beginning of the book (Figure 4). This book emphasizes three blessings within the human realm: firstly, the existence of human relationships within the vast world; secondly, the multitude of laws within these relationships; thirdly, understanding these principles leads to personal fulfillment. These three blessings are also known as the "three lives of human relationships" (Jinrin no sanshō 人倫の三生) (Furuya 1921b, p. 37). The book asserts that ignorance of human relationships makes one no different from beasts and inanimate objects. Furthermore, "human relationships can be subdivided into seven forms of grace, including grace from heaven and earth, grace from the king, grace from parents, grace from teachers, grace from friends, grace from relatives, and grace from all sentient beings. Among these, the second and third forms of grace are immeasurable, deeper than the vast

sea or the towering Mount Sumeru" ([Furuya 1921b](#), p. 115). While the direct sources for these discussions on human relationships cannot be identified, they are evidently linked to the flourishing background of Zhuism during the Edo period. The compiler, Isai, likely intended to blend Buddhist and Confucian ideologies to promote virtuous conduct and discourage wrongdoing (Kanzen chōaku 勧善懲悪).

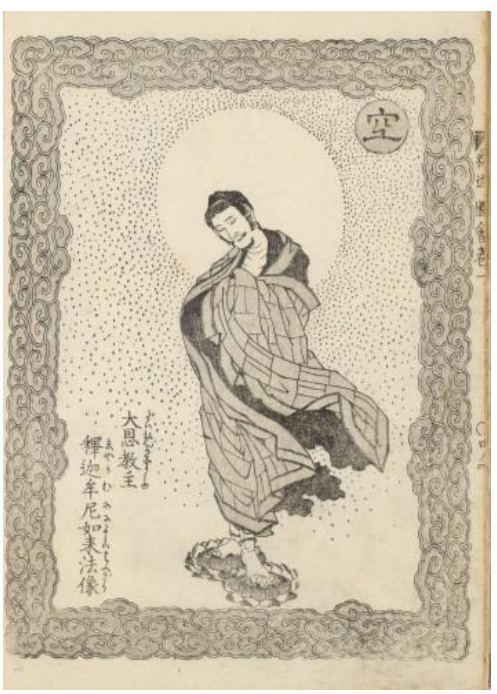

**Figure 4.** *Shaka goichidaiki zue* Volume 1. Caption: "The Great Grace Master, Shakyamuni Tathagata Dharma's Image". Image cited from: [https://kokusho.nijl.ac.jp/biblio/100389889/6?ln=ja](https://kokusho.nijl.ac.jp/biblio/100389889/6?ln=ja) (accessed on 3 January 2024) © Kobe University Library, Japan.

This work particularly extols the kindness of rulers and parents, repeatedly citing multiple stories to emphasize their significance. For instance, when the three brothers of Prince Sudhdhodhana inherited surrounding territories outside of Kapilavastu, they expressed gratitude to King Sihahanu for his grace. After offering his daughters to King Sudhdhodhana, Minister Suprabuddha was appointed as a regional lord, and he also expressed deep gratitude for the grace from the king.

The emphasis on parental grace becomes the focal point of this work. Although Prince Siddhartha did not accede to the throne, as desired by his father, instead leaving the royal palace to pursue a monastic life, this book repeatedly emphasizes his inability to forget the kindness of his parents. Importantly, this does not hinder his spiritual pursuit; rather, it portrays Shakyamuni as more compassionate, making him more relatable to the people of Japan during the Edo period. The book directly quotes the Difficulty of Repaying the Profound Kindness of Parents Sutra 父母恩重経 four times, an apocryphal sutra developed in China that fused Confucian ethical principles with Buddhist teachings on filial piety. Before the birth of Siddhartha, Maya dreamt of his teaching, elaborating on the ten great graces a mother bestows upon her children, including protecting the child in the womb and enduring pain during childbirth. After his enlightenment, Shakyamuni also preached this sutra multiple times to his father, King Sudhdhodhana, and his stepmother, Gautami. Learning of King Sudhdhodhana's grave illness, Shakyamuni swiftly led a group of monks to visit him, offering teachings to alleviate his suffering. After the king's passing, Shakyamuni led a group to cremate his father's body and bury him beside him next to the grave of Maya on the Sunset Hill (Sekiyōsan 夕陽山), not only portraying Shakyamuni as a filial son but also conveying an exemplary spousal relationship.

The relationship between Shakyamuni and his father resonates with the relationship between Shakyamuni and his son, which has been highly valued in Japanese literature. For example, in the *Shaka no Honji*, there is an episode where the young prince Siddhartha expresses sorrow at not having a mother while observing a bird and its offspring. In some texts, this episode has then been transposed onto the king and his son, Rahula. In the *Shaka goichidaiki zue*, Yasodhara preserved a sleeve left by Prince Siddhartha and handed it over to Rahula at the age of 11, as a memento (Katami 形見), urging him to seek his father. This episode, found in the *Sutra of the Collected Stories of the Buddha's Deeds in Past Lives*, albeit borrowed, is presented in a manner distinct to Japanese sensibilities, evoking profound emotions.

The marital relationship between Prince Siddhartha and Yasodhara is also noteworthy. According to the book, King Sudhdhodhana initially selected two exceptional beauties for Prince Siddhartha, but he showed no interest. Later, Yasodhara appeared, and Prince Siddhartha smiled at her and gifted her a coral-made ornament. Yasodhara declined, asserting that her admiration was based on Prince Siddhartha's virtues rather than any desire for material possessions. Prince Siddhartha was delighted and praised her for possessing women's virtues (Futoku 婦徳) ([Furuya 1921b](#), p. 91). When Siddhartha finally left the palace, he pointed at Yasodhara's abdomen, predicting that she would conceive three years later. The three-year period exceeded the typical gestation period of ten months, raising doubts about Yasodhara's chastity and subjecting her to gossip. However, she endured silently, obeying her husband's command, and successfully gave birth to Rahula, raising him to adulthood. This portrayal of the couple is influenced by feudalistic master–servant relationships, reflecting the limitations of that era.

Meanwhile, the book depicts their affection and subsequent separation in a unique way. In the *Sutra of the Cause and Effect*, the celestial beings from Pure Abode and Desire Heaven filled the sky, causing people to fall asleep through their divine powers and naturally opening the north gate of the palace, aiding Prince Siddhartha's renunciation. In traditional images of Buddha's life, Prince Siddhartha, riding an adorned horse, leaps over the castle wall on a cloud emerging from under his feet. Sometimes, these depictions also include the Four Heavenly Kings supporting the horse's legs to muffle its hoofbeats. Then, chaos ensued within the palace, as everyone drummed and shouted due to the absence of Prince Siddhartha. On the contrary, in the *Shaka goichidaiki zue*, there is no divine intervention during Prince Siddhartha's renunciation. Instead, Prince Siddhartha accomplished his renunciation with Yasodhara's help. King Sudhdhodhana constantly organized festivities in the palace to dissuade Siddhartha from renouncing worldly life. Numerous attendants encircled Prince Siddhartha, preventing his escape. One night, Prince Siddhartha seized the opportunity to leave the royal palace, as Yasodhara managed to ensure that the attendants were all asleep. Traditional images of Buddha's renunciation depicted the empty palace from the outside, while this book unexpectedly presents a scene from inside the palace, portraying Siddhartha's determination to leave (Figure 5). With the door left ajar, the night breeze blows in, the upper garment of a woman, Yasodhara, hangs limply, and a maid lies asleep. The left side of the depiction features Prince Siddhartha embarking on the journey of enlightenment amidst the bright daylight, while on the right side lies the pitch-dark royal palace, creating a stark contrast between the two. Siddhartha's departure was an act of determination and resolve for himself, praised by celestial beings, yet it dealt a tremendous blow to the people remaining in the palace. Hokusai's stark depiction of the emptiness within the palace reflects the emptiness in the hearts of the people left behind.

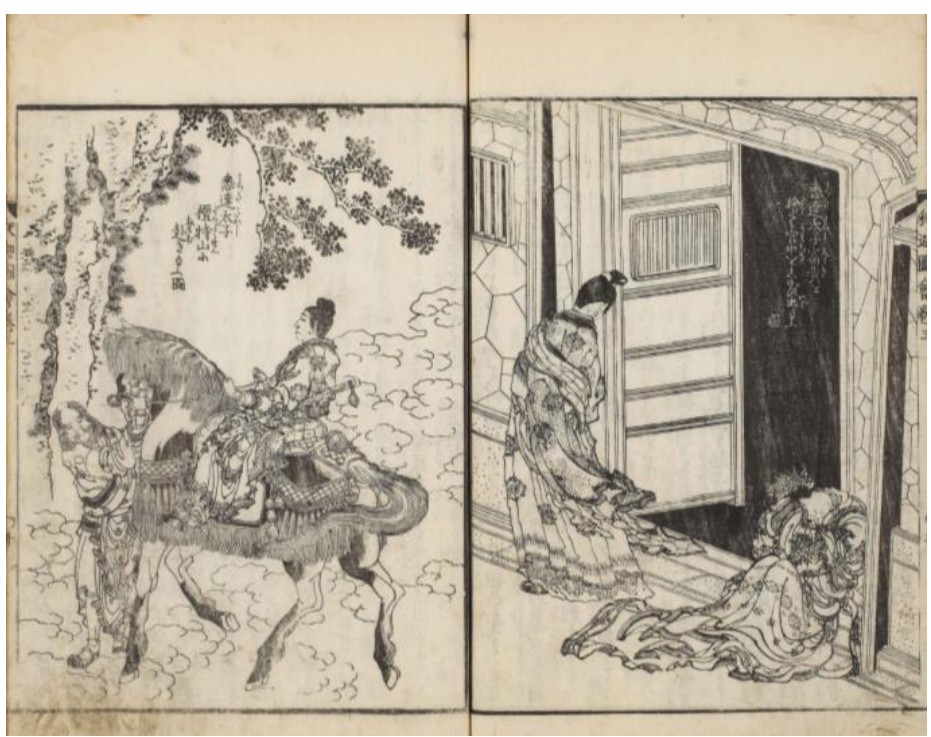

**Figure 5.** *Shaka goichidaiki zue* Volume 3. Caption: "Prince Siddhartha proceeds to the Mount Dandaloka (**Left** image). Prince Siddhartha renounced wealth and luxury, clandestinely departing from the palace (**Right** image)". Image cited from: https://kokusho.nijl.ac.jp/biblio/100389889/93?ln=ja (accessed on 3 January 2024) © Kobe University Library, Japan.

This book also portrays the teacher–student relationship. Shakyamuni sought guidance from many sages, including Alara Kalama. According to the narrative of this book, Shakyamuni was tasked with picking vegetables and fetching water. Due to his inability to discern profound truths through these ordinary tasks, he incurred the rebuke of Alara Kalama, who disciplined him with thirty strikes of a staff. Traditional depictions of ascetic practices in Buddhist images typically portray scenes of Shakyamuni drawing water from rivers in a simplified manner. These scenes are often small in scale, with symbolic representations that fail to convey the nuanced circumstances. In contrast, the illustration in this book spans two pages and vividly portrays Shakyamuni being struck by the staff of Alara Kalama. Buckets and baskets are placed next to Shakyamuni, who is lying on the ground. This illustration is a creative endeavor attributed to Hokusai after familiarizing himself with the textual descriptions. Notably, within the Buddhist narratives, Shakyamuni Buddha is always portrayed as calmly overcoming adversities. Instances of Shakyamuni being struck, as depicted in Figure 6, could not be found in other representations. This illustration resonates as a compelling scene, evoking a profound emotional response even from those who may not adhere to Buddhist beliefs. At first glance, this illustration may appear somewhat disrespectful toward Shakyamuni, and it might even be perceived as a form of blasphemy by certain extremist religious individuals. However, it conveys the relatability of Shakyamuni as an ordinary individual who, like those around the reader, necessitates the reverence of teachers and dedication to the enlightenment.

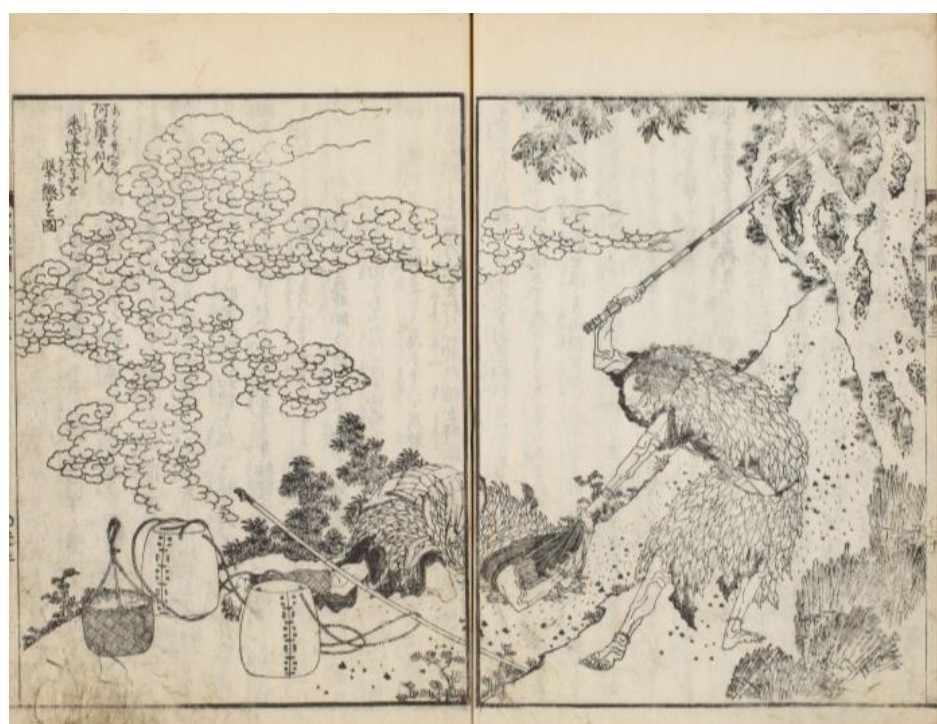

**Figure 6.** *Shaka goichidaiki zue* Volume 3. Caption: "Sage Alara disciplined Prince Siddhartha". Image cited from: https://kokusho.nijl.ac.jp/biblio/100389889/105?ln=ja (accessed on 3 January 2024) © Kobe University Library, Japan.

In addition to promoting the virtues of human relationships, this book also portrays the consequences of going against natural and moral principles through various negative examples. For example, Shakyamuni tells the story of King Vidudabha's past life as a big fish that was eaten by villagers of the Sakiya Kingdom, explaining the inevitability of King Vidudabha's invasion and the destruction of the Sakiya Kingdom through the concept of cause and effect. At the same time, for the cruel King Vidudabha, the book emphasizes that although he may prosper for a while, retribution is inevitable. When the time comes, punishment will be inflicted. The illustration selects a scene of punishment by the thunder god, which is highly impactful, to emphasize that actions contrary to divine favor and moral principles will undoubtedly face the consequences. The bold composition of the scene contrasts the thunder god, wielding weapons and striking down thunderbolts on the left, with the destruction of King Vidudabha's palace in the swirling blast on the right. The depicted moment is remarkably compelling, similar to the performance style *Mie wo kiru* (見得を切る)[6] in Kabuki theater. Besides, contrary to the traditional depiction of the thunder god, Hokusai boldly contrasts the scene by drawing thunder drums behind, replacing the clouds with a thunder beast, and portraying their faces with stern and fearsome expressions. It can be said that the thunder god here, to some extent, becomes a representation of the divine way and moral principles.

## 4. Reprints during the Meiji Period

The impact of the *Shaka goichidaiki zue* extended beyond the Edo period and experienced a resurgence in reprints between the fifteenth and twenty-fifth years of Meiji (1882–1892), garnering renewed attention from the public. As shown in Table 1, at least ten distinct editions reprinted by various publishers during this decade have been identified. Although their titles differ slightly, such as the omission of the character "Ki 記" or a change in the book title to *Shakuson gojitsudenki* 釈尊御実伝記, the textual content of these publications remains consistent with the *Shaka goichidaiki zue* compiled by Ishisai in the late Edo period.

**Table 1.** Reprints during the Meiji period.

| Title | Publisher | Year |
|---|---|---|
| *Shakuson goichidai zue* 釈迦御一代図会 | Nissinkan 日新館 | 1882 |
| *Shakuson goichidai zue* 釈迦御一代図会 | Yanagihara tomoshichi 柳原友七 | 1884 |
| *Shakuson goichidai zue* 釈迦御一代図会 | Aoki tsunesaburō 青木恒三郎 | 1884 |
| *Shaka nyorai ichidaiki zue* 釈迦如来一代記図会 | Fūgetsudō 風月堂 | 1885 |
| *Shakuson gojitsudenki* 釈尊御実伝記 | Kakuseisha 鶴声社 | 1887 |
| *Shakuson goichidai zue* 釈迦御一代図会 | Mori senkichi 森仙吉 | 1887 |
| *Shakuson gojitsudenki* 釈尊御実伝記 | Monko Heikichi 門戸平吉 | 1889 |
| *Shakuson gojitsudenki* 釈尊御実伝記 | Nakamura Yoshimatsu 中村芳松 | 1889 |
| *Shaka gojitsudenki* 釈迦御実伝記 | Koishikawa shuppan 礫川出版 | 1890 |
| *Shakuson goichidai zue* 釈迦御一代図会 | Matumaeya shokan 松前屋書館 | 1892 |
| *Shakuson goichidai zue* 釈迦御一代図会 | Kimura jūsaburō 木村重三郎 | 1893 |

In terms of imagery, the covers of these reprinted books vary, and the illustrations within them largely adhere to the pattern of Katsushika Hokusai's illustrations, albeit with minor differences in detail. Compared with Figure 7, the two Meiji-era reprinted illustrations below, while maintaining a basic consistency in terms of composition, exhibit noticeable deficiencies in artistic expression. Due to the necessity of reusing lead plates (rather than wooden ones) for engraving, not only have there been changes in the position of the thunder god, but more importantly, the lines lack smoothness. Details such as the depiction of stormy whirlpools and scattered fragments are markedly different, with a diminished visual impact. Furthermore, in terms of the relationship between text and image, Figure 8b largely adheres to the "word within the picture" (Gathūshi 画中詞) pattern of Figure 7, where text serves a supplementary explanatory role. In contrast, Figure 8a incorporates a substantial amount of text, resulting in a scenario where the text takes precedence over the picture.

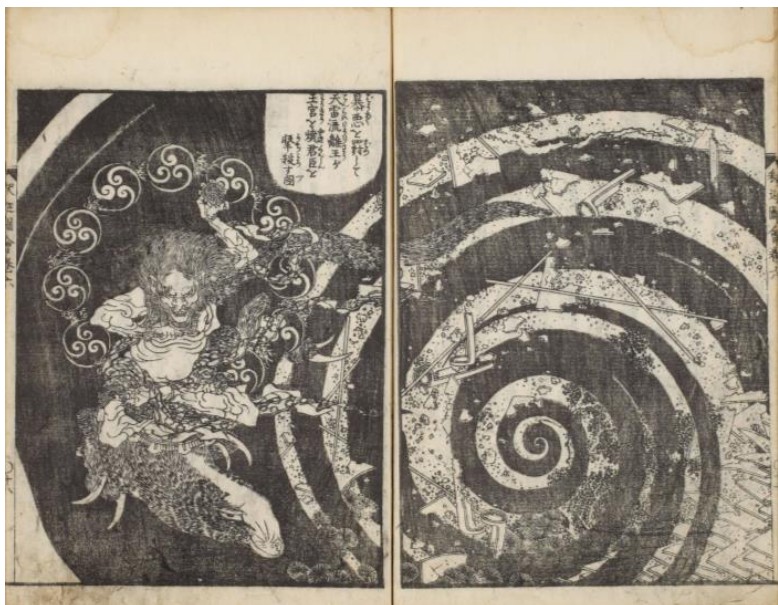

**Figure 7.** *Shaka goichidaiki zue* Volume 6. Caption: "The celestial thunder burned down the palace of King Vidudabha, slaying rulers and ministers as a punishment for their tyranny". Image cited from: https://kokusho.nijl.ac.jp/biblio/100389889/212?ln=ja (accessed on 3 January 2024) © Kobe University Library, Japan.

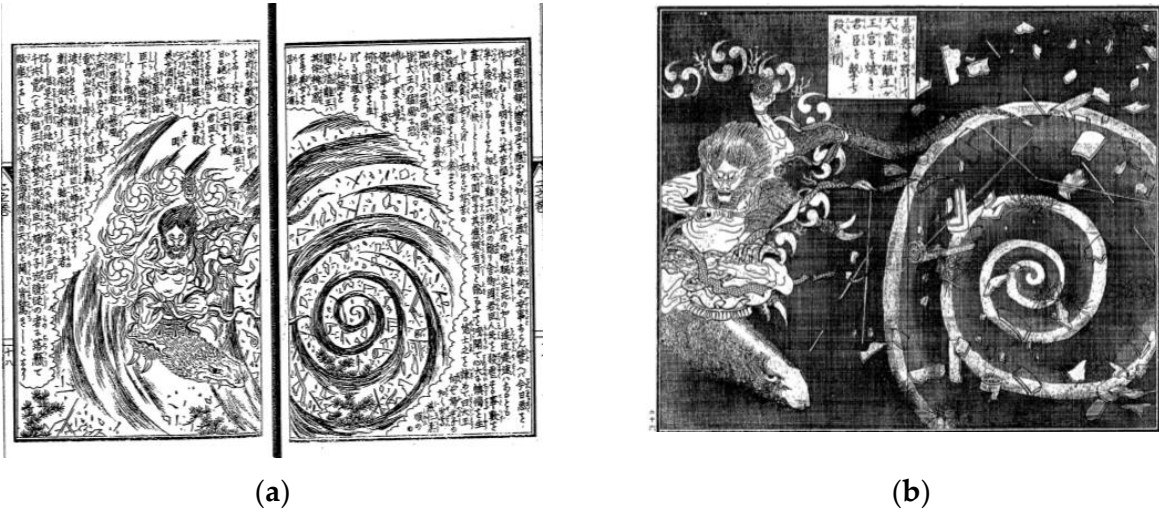

**Figure 8.** (**a**) *Shaka nyorai ichidaiki zue*, published by Fūgetsudō, 1885. Image cited from: https://dl.ndl.go.jp/pid/816586/1/19 (accessed on 3 January 2024) © National Diet Library, Japan. (**b**) *Shakuson goichidai zue*, published by Kimura jūsaburō, 1893. Image cited from: https://dl.ndl.go.jp/pid/816571/1/187 (accessed on 3 January 2024) © National Diet Library, Japan.

Curiously, why did a sudden wave of reprints emerge during this period? The reprinted books included another preface after the initial preface inscribed by Sōgen, stating: "The record of the Buddha's life in this book (Shaka goichidaiki zue), is the most detailed, yet currently challenging to obtain. Therefore, in this reprint, no single sentence is omitted from the textual content, and illustrations are engraved using lead plates. It is hoped that sentient beings with affinity and virtuous individuals (Kunshi 君子) in society will together propagate the immeasurable and vast merits of the Tathagata" (Monko 1889, p. 2).

The final sentence of the preface particularly urges the collective promotion of Buddhism. This appeal arose due to the anti-Buddhist movement Haibutsukishaku 廃仏毀釈 in the early Meiji era, prompting Japanese society to reassess and evaluate Buddhism. In the *Shakamuniden*, Tetsujirō Inoue asserted, "Recounting accurate facts about Shakyamuni's deeds is exceedingly challenging. Posthumous reverence by later Buddhist believers has led to the inclusion of numerous supernatural and mysterious anecdotes. Consider, for instance, traditional biographies of Shakyamuni in Japan, which, while abundant, are largely extravagant and fictional narratives. Careful consideration is necessary in selecting and discarding. Recently, the study of Buddhism in the Western world has gradually gained momentum, presenting noteworthy documents related to Shakyamuni's deeds. If this book can introduce the truth of a globally renowned figure buried within extravagant and fictional tales, it would undoubtedly be a significant achievement" (Inoue 1902, pp. 2–3). The traditional Buddhist biography, since the Chinese translation of Buddhist scriptures, is perceived as "fictional tales" containing "supernatural and mysterious anecdotes", contrasting with Western studies while striving to unveil the "truth of a globally acclaimed figure" hidden within these "fictional tales". Despite wavering in reliance on narrativity due to Western influence, there seems to be an ongoing attempt to derive new significance from this narrative approach. Additionally, it is notable that the recognition of "Buddhist biographical literature" as "tales" has already been established, indicating a modern shift in portraying the image of Shakyamuni, approaching him not merely as a "human-being" but as a "personality".

Further considering the preceding preface, the primary reasons facilitating the reprinting of this book can be outlined as follows. Firstly, in terms of content, this book provides the most detailed and objective records. Despite retaining numerous mysterious anecdotes, as previously mentioned, the deification of Shakyamuni is notably subdued, rendering it more comprehensible for the general populace. Moreover, the book extensively cites

Buddhist scriptures and related literature, providing readers with a sense akin to modern historical research. Perhaps for this reason, some reprints deliberately changed the title from "ichidaiki" (personal biography) to "jitsuden" (factual biography), emphasizing that Yamada Isai's Japanized adaptation is the most objective and trustworthy. Consequently, publishers during the Meiji period insisted on maintaining the content without alteration. Secondly, the book's indigenization of thought facilitated a stronger connection with sentient beings, including those living in the world and adherents of Confucianism known as Kunshi (gentlemen). As previously mentioned, this book, integrating Japan's contemporary situation, literary traditions, and Edo-period Confucian thought, adapted ethical relationships, such as sovereign and subject, parent and child, husband and wife, and teacher and student, to align with Japanese circumstances, particularly suitable for the Edo populace deeply influenced by Zhuism. In the Meiji era, the book's ideas did not appear significantly outdated. For example, Shakyamuni is depicted as born into a royal family boasting immense power. This is within the narrative work but has a direct connection to real-world royal authority. As Umehara Kunzan's *Shakamunibutsu* emphasizes, Yamada Isai depicts the scene of the imperial capital, subtly praising the virtues of King Sudhdhodhana, celebrating the prosperity of the era and the flourishing of the nation through his words (Umehara 1894, p. 38). Sueki Fumihiko further pointed out that the resurgence of interest in Shakyamuni within Buddhism aligns parallelly with the revival of ancient myths in Kokugaku 国学 (Japanese national study) and Fukko Shintō 復古神道 (Restoration Shinto). In the modern era, as Buddhism actively embraced the absolute monarchy of the imperial state and developed under this premise, it is not unrelated to the implicit structural framework of the Wheel-Turning King plus Buddha (Sueki 2018, p. 17). Thus, it is not difficult to understand why "Shaka gojitsudenki" was brought to the forefront during this period, serving as a vigorous Japanese-style promotion of Buddha's biography.

## 5. Conclusions

The *Shaka goichidaiki zue* is a meticulously crafted text with carefully illustrated graphics, executed near the end of both Isai and Hokusai's lives. Due to its delicate integration of this-worldly elements and other-worldly components, the book gained tremendous sales upon its publication in Edo, Osaka, and Kyoto. Interestingly, in the late 19th century, amidst changing perceptions of Buddha, this book experienced a revival and vigorous promotion, resulting in the reprinting of no fewer than ten editions. This resurgence was primarily due to the book's localized adaptation concerning royal authority, literary traditions, ethical relationships, and ultimately, the creation of a Buddha that resonated with the ideological needs of the Japanese.

Tsuji Zennosuke once proposed the theory of the Kinsei bukkyō darakuron 近世仏教堕落論 (decline of early modern Buddhism), asserting that Buddhism during the Edo period became more superficial and morally declined compared to other eras (Tsuji 1931, p. 516). However, in recent years, Japanese scholars, such as Ryō Nishimura and Sueki Fumimichi, have highlighted the necessity to reevaluate Japan's early modern Buddhism and have endeavored to establish a new historical context (Nishimura 2018; Sueki 2023). This article contends that the intellectual landscape of early modern Buddhist thought exhibits multifaceted complexity. In the Edo period, Kokugaku was dedicated to reconstructing and personifying Japanese mythology based on the *Kojiki* and *Nihon Shoki*, in order to pursue a pure ideology that excluded the influences of foreign religions, such as Buddhism and Confucianism. Fukko Shintō holds that the Zōka sanshin 造化三神 (Ame-no-Minaka-Nushino-Kami, Takamimusubi-no-Kami, and Kamimusubi-no-Kami) performed the creation of all things before the descent of the heavenly descendants. These three deities are both the supreme deities in mythology and possess a profound character as ethnic deities. In contrast, early modern Buddhism exhibits a different posture, actively absorbing Confucian ethics, royal authority, and this-worldly elements, with the aim of creating a Japanese Buddha's story. While the phenomenon of popularization of Buddhism is evident in the Edo period, it does not signify a loss of vitality or creativity. The *Shaka goichidaiki zue* effectively

presents the new dimensions of Buddhist literature and art in terms of vernacularization and visualization. In other words, it is necessary to reexamine the position of Buddhism not only in terms of doctrinal teachings but also considering literature, images, and the overall intellectual landscape. This involves questioning the relationship between Buddhism and various media, and exchanges of different religious ideologies.

Lastly, this article aims to transcend the often-segmented views between the modern and pre-modern eras, serving as part of an effort to bridge the classical and modern periods. It is perhaps unforeseen that the *Shaka goichidaiki zue*, heritage from the pre-modern era, played a role in the process of modernization of Buddhism during the late 19th century.

**Funding:** This research was funded by the Humanities and Social Science Fund of Ministry of Education of China, grant number 22JJD750002. The research project is entitled "Eastern Literature and Civilization Exchanges: A Comparative Study of Multilingual Ancient Eastern Literature Illustrated Books 东方文学与文明互鉴:多语种古代东方文学插图本比较研究".

**Data Availability Statement:** No new data were created or analyzed in this study. Data sharing is not applicable to this article.

**Acknowledgments:** I would like to express my gratitude to Kazuaki Komine for sharing valuable material and insightful suggestions.

**Conflicts of Interest:** The author declares no conflicts of interest.

## Notes

[1] The book is woodblock print, ink on paper, and its dimensions are 24.9 × 17.8 cm.

[2] "Hokusai" is an abbreviation of the original name "Hokusai Tokimasa 北斎辰政", inspired by the reverence of the Hokushin Myoken Bodhisattva 北辰妙見菩薩 within the Nichiren sect, which deified the North Star and the Big Dipper.

[3] *Hokusai sensei gafu* (Accession Number 11.46051), William Sturgis Bigelow Collection, Boston Museum of Fine Arts. Available online: https://collections.mfa.org/objects/700511/hokusai-sensei-gafu (accessed on 3 January 2024). Additionally, the Boston Museum of Fine Arts also houses some replicas (Accession Numbers 11.46048 and 11.46049) of the *Shaka goichidaiki zue*. For instance, Narui Sadao 成井貞央 meticulously copied the scene of "The Founding of the Jetavana Temple" from this book. The title *Dairokugo Hokusai no zu 第六号 北斎之図* indicates that this album of sketches was one of a set of at least six. It was copied from the printed book illustration rather than the preliminary drawings. Evidentlly, the illustrations in the *Shaka goichidaiki zue* are often considered prototypes by artists, demonstrating a significant artistic influence.

[4] The Arthur Tress Collection of Japanese Illustrated Books, Kislak Center, University of Pennsylvania Libraries. Available online: https://web.sas.upenn.edu/tressjapanese/2021/02/09/yamada-isai-%E5%B1%B1%E7%94%B0%E6%84%8F%E6%96%8E-and-katsushika-hokusai-%E8%91%9B%E9%A3%BE%E5%8C%97%E6%96%8E-shaka-go-ishidaiki-zue-%E9%87%88%E8%BF%A6%E5%BE%A1%E4%B8%80%E4%BB%A3%E8%A8%98%E5%9B%B3/ (accessed on 3 January 2024).

[5] According to the book, the other two things that females need to be cautious about are arrogance and sleep.

[6] In Kabuki theater, actors momentarily freeze their movements, creating a fixed image to convey a dramatic climax and the heightened emotions of the character.

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
