# Peer review of "Shaka Goichidaiki Zue: Vernacularization and Visualization of Buddha’s Biography in Nineteenth-Century Japan"

_religions, doi:10.3390/rel15010099_

Round 1
Reviewer 1 Report
Comments and Suggestions for Authors
This is a well written paper that was highly interesting to read discussing the nature of the text Shaka Goichidaiki Zue. The argument that this text achieved success due to its balancing of religious and secular aspect, and its revival in the 19th century by adapting the creation of a Buddha that resonated with the ideological times is well made. So is the point that the Shaka Goichidaiki Zue presents new dimensions of Buddhist literature and art in terms of vernacularization and visualisation, and the need to consider the overall intellectual landscape. The following comments are concerned with the use of the modern conceptualisation of the 'religious' and 'secular' for framing the analysis without acknowledging the complications of this and debates around the use of such concepts in the Japanese context.
Some comments for reflecting upon the religious-secular binary applied as form for analysis:
The author will undoubtedly be familiar with the debates within so-called critical religion that in the context of Japan looks at the modern importation of the binary classification of religion and the secular. The overall argument would be enhanced by the author using these concepts in a less taken for granted way, reflecting more critically of the use of these modernist framings, which function also in this context in ideological ways.
Examples:
Line 56-57 : The author states the text achieved a striking balance between religious and secular aspects, but without reflecting upon their own modernist imposed framing into which the text is placed. I would suggest to elaborate on "How the religious/secular binary is applied to an analysis which point to a non-binary complexity?" It seems somewhat contradictory to make this point given exactly the fact that the author emphasises that it is unadvisable to put the text into a binary opposition of pre-modern Buddhist narratives. This is particularly poignant as the author goes on to point to the "religious" aspects of the "zue" itself.
In my view, the discussion continues in the same way such as in Line 101 and 123 to use the modern concept of religion in too unreflective a manner. I would suggest to clarify or include a discussion of the modernist secular-religious binary since this conceptualisation frames the main argument.
More clarification of these debates in critical religion would enhance the discussion also when the author states in Line 298-299 how the religious background is secularised.
Not including such debates gives rise to some confusion also around Line 328 when it seems the "religious" is equated with the fantastical. Would it be more appropriate to use terminology of 'this worldly' or 'other worldly' rather than secular and religious? Secular here seems to used in the text more accurately pointing to social relations.
Line 425: This point made needs further elaboration. Also, Ikeda Daisaku has of course written much about Nichiren but his work may not be regarded as authoritative, the point made would be more solid if also backed up by other scholarly work making such points.
Also, it is worth pointing out that in the next sentence, from Line 426, that Nichiren does not, as far as I know, talk about females in the way presented in the Shaka Goichidaiki Zue. This needs to be clarified. Here the author seems to miss an opportunity to point out the change in narrative to a gendered one. This is very interesting and could be further explored.
Line 663 Capital letter at beginning of sentence missing
Author Response
Please see the attachment(Cover Letter).

Reviewer 2 Report
Comments and Suggestions for Authors
Let me congratulate the author on an excellent article that calls attention to an important and overlooked work that tells us a great deal about Japanese Buddhism in the Nineteenth Century. So much of what we assume about the period—and this is true for Nineteenth- Century China as well—relies on assumptions that may be outdated and that deserve careful reconsideration. In the popular imagination, Hokusai is often associated with the growing secularization of Japanese society, and his most famous works of the late period include the world-famous series One Hundred Views of Mt. Fuji. This article complicates not only our understanding of Hokusai but also of Japanese modernization, to which Buddhists responded more creatively, it would seem, than we might have assumed. I believe that this article will be of broad interest to East Asian historians and art historians as well as to scholars in religious studies.
Author Response

(The authors gave the same response as above.)

Reviewer 3 Report
Comments and Suggestions for Authors
This paper brings new text and new idea about Buddha's biography in some extance. Shaka goichidaiki zue 釈迦御一代記図会 is unique in world's Buddha's biography across in premodern and modern Asia. It shows the tradition of Buddhist literature and Japanese Vernacularization and Visualization which show the way of modernization for other countries.
The reception and recreation in the Shaka goichidaiki zue shows a common rule during the cultural transmission. Research on this reminds us to respect and integration the local culture to creat a diverse world culture. So that I fully support the publication of this paper.
What i suggest is that, the author might emphasis the social background of Japan at that period, in abstract and conclusion. Foe example, Kokugaku 国学(Japanese national study ) , Fukko Shintō 復古神道 (Restoration Shinto),Kinsei bukkyō darakuron 近世仏教堕落論(decline of early modern Buddhism) are important and valuable thoughts for us to understand the history of buddhist literature in Edo and Meiji period. If the paper discussed more in conclusion, readers would understand more about the buddhist culture in Japan and world.
Author Response

(The authors gave the same response as above.)
